

# EPypes: a framework for building event-driven data processing pipelines

Oleksandr Semeniuta[1] and Petter Falkman[2]

[1] Department of Manufacturing and Civil Engineering, NTNU Norwegian University of Science and Technology, Gjøvik, Norway
[2] Department of Electrical Engineering, Chalmers University of Technology, Gothenburg, Sweden

## ABSTRACT

Many data processing systems are naturally modeled as pipelines, where data flows though a network of computational procedures. This representation is particularly suitable for computer vision algorithms, which in most cases possess complex logic and a big number of parameters to tune. In addition, online vision systems, such as those in the industrial automation context, have to communicate with other distributed nodes. When developing a vision system, one normally proceeds from ad hoc experimentation and prototyping to highly structured system integration. The early stages of this continuum are characterized with the challenges of developing a feasible algorithm, while the latter deal with composing the vision function with other components in a networked environment. In between, one strives to manage the complexity of the developed system, as well as to preserve existing knowledge. To tackle these challenges, this paper presents EPypes, an architecture and Python-based software framework for developing vision algorithms in a form of computational graphs and their integration with distributed systems based on publish-subscribe communication. EPypes facilitates flexibility of algorithm prototyping, as well as provides a structured approach to managing algorithm logic and exposing the developed pipelines as a part of online systems.

Corresponding author
Oleksandr Semeniuta,
oleksandr.semeniuta@ntnu.no

## INTRODUCTION

In recent years, the increased availability of computational resources, coupled with the advances in machine learning methods and ability to gather large amounts of data, opened new possibilities of developing more advanced data-driven systems. Visual data, acquired by various types of imaging equipment, constitutes one of the main inputs to advanced data analysis algorithms.

In manufacturing automation, vision systems has a long history of use in combination with dedicated automated equipment and industrial robots, serving a role of contact-less sensing for, amongst others, quality inspection and robot guidance. What differentiates industrial vision solutions from general-purpose computer vision systems, is their coupling with the associated mechatronic components possessing an actuation function. This entails that most industrial vision systems operate in online mode, with their operation being synchronized with external systems by various forms of remote communication.

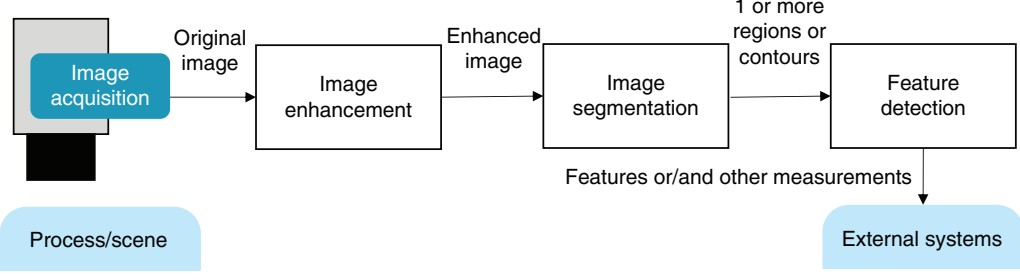

Figure 1  **Common steps of a vision pipeline.**       

When a new robotic system with vision sensing is developed, the early-stage system prototyping favors flexible tools and techniques that allow for iterating toward a functional solution quickly. When it comes to computer vision prototyping, the tools of the trade include OpenCV, as well as libraries from the Python data science ecosystem, most notably NumPy, SciPy, Pandas, Scikit-learn, Scikit-image, and others. Vision algorithm development is a challenging task in itself, as it requires a great deal of experimentation and tuning of numerous parameters and thresholds. Another challenge with early-stage prototyping of vision algorithms to be used with robotics and automation solutions is their coupling to other networked components. Establishing communication interfaces can be time consuming, and is often done as a patchwork, which is difficult to maintain.

Many data processing systems can be logically modeled as direct graphs in which data is being gradually processed by the computational nodes. This is particularly characteristic of vision systems: after image capture and acquisition, an input image is obtained in memory and fed to a series of transformations leading to the application-specific output. Such pipeline can be comprised of the steps of image enhancement, image segmentation, and feature detection (Fig. 1).

This idea has been formalized with the abstract concept of data flow, and has found its application in many areas, including distributed data processing, machine learning, embedded software development, and digital signal processing. MATLAB Simulink and LabVIEW are the traditional engineering tools whose programming model is based on data flow. In data engineering and data science areas, tools like Apache Storm, Apache Airflow, Luigi, and Dask employ explicit data flow construction and execution. Needless to mention that the deep learning libraries, such as TensorFlow, Caffe, and Theano, construct and train models as directed acyclic graphs (DAGs).

This paper tackles the problems of both (1) vision algorithms development and (2) their integration into distributed environments. This is done by introducing EPypes, a Python library[1] for construction and execution of computational graphs, with the built-in capability of exposing the graphs as reactive pipelines. The latter are intended to be a part of publish-subscribe systems. In addition to the software tools, this paper presents a system development method that facilitates transition from ad hoc prototyping phase to well-structured system integration phase without compromising the developers' flexibility.

[1] The EPypes implementation is available under the 3-clause BSD license at https://github.com/semeniuta/EPypes.

The practical applicability of the proposed framework is validated in a distributed experimental setup comprised of a robot, an image acquisition service, and an image processing component, communicating in a publish-subscribe manner using ZeroMQ middleware. It is shown that the EPypes architecture facilitates seamless transition between various deployment configurations in a distributed computing environment.

This paper is structured as follows. First, the background areas are introduced, including overview of computational systems based on DAGs, the Python data science/computer vision ecosystem, and event-based middleware. Further, the EPypes abstractions are presented with code examples and architectural relationships. Finally, a distributed system experiment based on EPypes provides a more detailed view into the runtime properties of the framework.

## BACKGROUND

### Graph-based representation of computational systems

A wide range of computational systems, particularly those with streaming behavior, can be represented as directed graphs, in which data is routed through processing nodes. Not only is this representation accessible to human understanding (particularly for engineers), but it also has been used in various settings to realize improvement of the function of the systems.

Control engineering and signal processings has a long tradition of graphically modeling systems in a form of block diagrams. MATLAB Simulink and LabVIEW are widely used in this context as engineering tools with formally defined abstractions. The field of cyber-physical systems (CPS) makes great use of graph-based system models together with the associated models of computations (*Lee & Seshia, 2011*). A notable CPS modeling environment is Ptolemy II.

In computer science, graph-based representation of systems has been used for a range of different purposes: data flow models, task graphs (for parallel processing scheduling), symbolic representation of computational expressions (for machine learning and automatic computation of gradients), representation of concurrent process networks (e.g., Communicating Sequential Processes), workflow languages, etc. In the software engineering community, the *pipes and filters* architecture applies the same ideas to data processing systems design and development. The well-known pipes mechanism of Unix-like operating systems has proved to be particularly powerful when it comes to composition of multiple tools to solve a complex task.

Data science has seen a surge of tools based on explicit handling of data processing systems in a form of DAG. Many of them are intended to be run on a computing cluster, and the DAG architecture in this case facilitates scheduling of parallel execution of data processing tasks. Apache Storm is a cluster-based stream processing engine. Apache Airflow is workflow management platform for batch processing on a cluster. Dask is a Python parallelization library that utilizes DAG modeling for scaling algorithms written with NumPy and Pandas primitives to be used with massive datasets.

## Python data science/computer vision ecosystem

The open source movement has gained a big popularity within the fields of data science, computer vision, and robotics in recent years. Even though the established proprietary engineering tools are pervasive in the industrial context, they often lack flexibility and hinder a deeper understanding of how a system functions. Conversely, open source tools provide community-contributed implementation of common functionality, which is flexible to use and allows for building more scalable and reproducible solutions.

In computer vision, the OpenCV library has become a de-facto standard providing a pool of community-contributed image processing and computer vision algorithms. Similarly, the point cloud library (PCL) provides open-source routines for point clouds processing. A multitude of tools from the Python ecosystem are widely used for data science and scientific computing. They are built upon the NumPy array library, and include Pandas, Scikit-learn, Scikit-image, and many others. The abovementioned OpenCV and PCL, as well as many other low-level tools, expose Python bindings, which makes it possible to perform rapid system developed with preserved high performance of the applied algorithms.

## Events and publish-subscribe middleware

An event-driven system is characterized by a discrete state space, where state transition happen on occurrence of events at sporadic time instants (*Cassandras & Lafortune, 2008*). In distributed systems, events are often embodied as messages sent over a network in a *publish-subscribe* communication system. Such messages can signalize a change of a system state (change event) or a notification from an observation (status event), expressed as a tuple with a timestamp and an application-specific descriptive parameters (*Hinze, Sachs & Buchmann, 2009*). Message-based middleware provides a unified set of communication and input/output capabilities in such *sense-respond* systems.

Middleware allows to decouple the communicating components by introducing message queuing, built-in address resolution (e.g., via handling logical addresses such as topic names), and usage of a common data serialization format (*Magnoni, 2015*).

The defining components of a particular middleware solution are the communication protocol (transport-level TCP and UDP, wire-level AMQP, ZeroMQ/ZMTP, MQTT), the communication styles (request/reply, publish/subscribe), and the data serialization method (typically realized with an interface definition language like Protobuf or Apache Thrift). Many middleware solutions are based on a central broker, for example, ActiveMQ and RabbitMQ. The additional hop through the broker adds a constant value to the communication latency (*Dworak et al., 2012*). ZeroMQ is an example of broker-less middleware, in which the message queuing logic runs locally within each communicating component (*ZeroMQ, 2008*).

## EPYPES

EPypes is a Python-based software framework that combines *pipes and filters* and *publish-subscribe* architectures. It allows to develop data processing pipelines, the behavior of which is defined by their response to events. EPypes defines a *computational graph*, which is

a static data structure modeling a data processing algorithm, abstractions used for execution of computational graphs, and a hierarchy of *pipelines*, which extend the algorithm logic defined with computational graphs to be a part of a publish-subscribe system.

## Computational graph

At the core of EPypes lies `CompGraph`, a data structure that models a data processing algorithm as a computational graph, that is, as a network of functions and data tokens. Formally, a `CompGraph` can be described as a bipartite DAG $G$:

$$G = (F, T, E)$$

where $F$ is a set of functions, $T$ is a set of data tokens, and $E$ is a set of directed edges between functions and tokens and vice-versa. The latter implies that edges of only the following two kinds are permitted: $(f, t_i)$, where $f \in F$, $t_i \in T$, and $(t_j, g)$, where $g \in F$, $t_j \in T$.

A function $f \in F$ is associated with a Python callable. A token $t \in T$ represents a data object of an arbitrary type. If function $f$ correspond to a callable with $m$ input parameters and $n$ outputs, it has to be connected to $n$ input and $m$ output tokens. Let $\text{In}_f \subset T$ denote the set of input tokens to $f$, and $\text{Out}_f \subset T$ denote the set of output tokens from $f$.

Functions in $G$ are uniquely identified by their string-based names. This allows to use the same Python callable several times in the computational graph.

Once a computational graph $G$ is constructed, and it conforms to the requirements of acyclicity, its execution can be scheduled. Topological sort of $G$ results in an order of vertices (functions and tokens) so that all the directed edges point from a vertex earlier in the order to a vertex later in the order. With invoking functions in this topological order, all the precedence constraints will be satisfied.

For many computational procedures, one can typically distinguish between parameters carrying the primary data entities and parameters that tune the procedure. In this paper, the former are referred to as *payload parameters*, and the latter as *hyperparameters*[2]. Thus, tokens belonging to these two parameter sets of function $f$ form the input parameter set of $f$: $\text{In}_f = P_f \cup H_f$. It is further presumed that all hyperparameter tokens are *frozen*, that is, given fixed values, during the construction of graph $G$. The set of non-frozen source tokens is referred to as *free source tokens*, and is used to provide input to the computational graph.

In the computational graph example shown in Fig. 2, rectangle vertices represent functions in the function set $F = \{f_1, f_2, f_3\}$, white circular vertices represent payload tokens, and gray circular vertices—hyperparameter tokens. In accordance with the previously defined notation, each function in $F$ has the following associated token sets:

| | | | |
|---|---|---|---|
| $f_1$ | $H_{f_1} = \{t_2, t_3\}$ | $P_{f_1} = \{t_1\}$ | $\text{Out}_{f_1} = \{t_4, t_5\}$ |
| $f_2$ | $H_{f_2} = \varnothing$ | $P_{f_2} = \{t_4\}$ | $\text{Out}_{f_2} = \{t_6\}$ |
| $f_3$ | $H_{f_3} = \{t_7\}$ | $P_{f_3} = \{t_5, t_6\}$ | $\text{Out}_{f_3} = \{t_8\}$ |

Token $t_1$ is the only free source token, and its value is required to perform a computation.

[2] The term *hyperparameters* is borrowed from machine learning, where it refers to parameters that characterize a particular algorithm, as opposed to model parameters. Semantics of hyperparameter tokens in this paper is similar, although the considered computational graphs can be used to model a wide variety of algorithms.

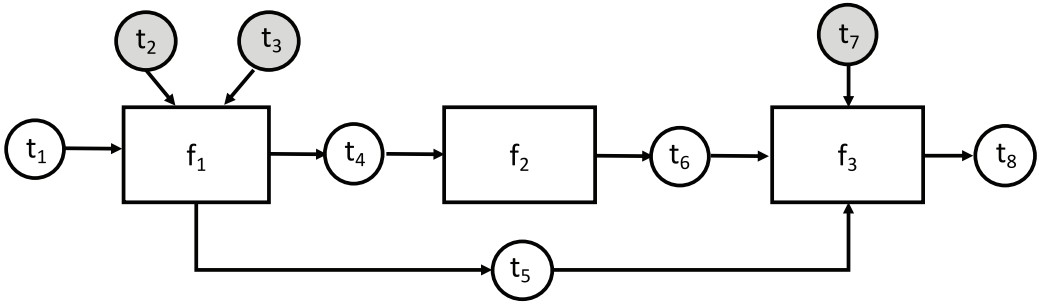

**Figure 2** **An example abstract computational graph.**

## A concrete example

Consider a simple computational graph that defines a processing chain in which a color image is first converted to grayscale, then blurred with a Gaussian kernel, with the blurred image further used to perform edge detection with the Canny algorithm.

The following listing shows the steps of the computational graph construction.

```python
import cv2

def grayscale (im):
    return cv2.cvtColor(im, cv2.COLOR_BGR2GRAY)

def gaussian_blur(img,kernel_size):
    return cv2.GaussianBlur(img, (kernel_size, kernel_size), 0)

func_dict = {
    'grayscale': grayscale,
    'canny': cv2.Canny,
    'blur': gaussian_blur
}

func_io = {
    'grayscale': ('image,' 'image_gray'),
    'blur': (('image_gray,' 'blur_kernel'),'image_blurred'),
    'canny':(('image_blurred,''canny_lo,' 'canny_hi'), 'edges'),
}

cg = CompGraph(func_dict, func_io)
```

After importing the OpenCV Python module (`cv2`), two helper functions are defined for grayscaling and blurring (the function for edge detection is used as-is). The structure of the computational graph is specified as two dictionaries. The `func_dict` dictionary defines mapping from unique function identifiers (in this case, strings "`grayscale`", "`blur`", "`canny`") to the respective callable objects. The `func_io`

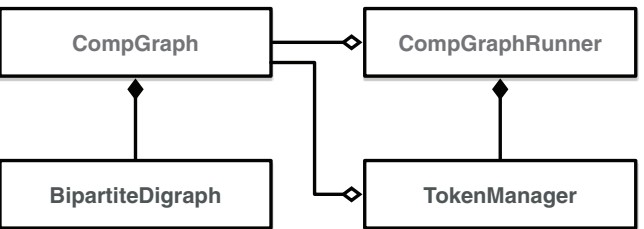

**Figure 3 Class digram of EPypes abstractions dealing with computational graphs.**

dictionary defines input/output relationships between the functions in a form of tokens. Each function identifier is mapped to a tuple describing input and output tokens that can be one of the following forms, depending on the respective functions' signatures:

- $(x, y)$ for single input and single output;
- $((x_1, ..., x_m), y)$ for multiple inputs and single output;
- $(x, (y_1, ..., y_n))$ for single input and multiple outputs;
- $((x_1, ..., x_m), (y_1, ..., y_n))$ for multiple inputs and multiple outputs.

An instance of `CompGraph` is then constructed based on `func_dict` and `func_io`.

To be executable, a computational graph has to be supplied to the constructor of `CompGraphRunner`. The latter is used to store the hyperparameter tokens and schedule execution of the graph with the topological sort. Internally `CompGraphRunner` delegates storage and retrieval of token data to an instance of `TokenManager` (Fig. 3).

In the following example, we specify the Gaussian blur kernel, and low/high threshold of the Canny algorithm in dictionary `params`. The latter, together with the original computational graph `cg` is used to construct a `CompGraphRunner`:

```
hparams = {
      'blur_kernel': 11,
      'canny_lo': 70,
      'canny_hi': 200
}

runner = CompGraphRunner(cg, hparams)
```

Visualization of this parametrized computational graph is shown in Fig. 4. The hyperparameter tokens are highlighted in gray.

To run a `CompGraphRunner`, its `run` method is invoked with keyword arguments corresponding to names and values of free source tokens. In the provided example the only free source token is `image`. Therefore, the running syntax is the following:

```
im = cv2.imread('image.jpg,'cv2.IMREAD_COLOR)
runner.run(image=im)
```

A `CompGraphRunner` can be used as a namespace for accessing any token value by the token key. The interface for this operation is the same as for a Python dictionary.

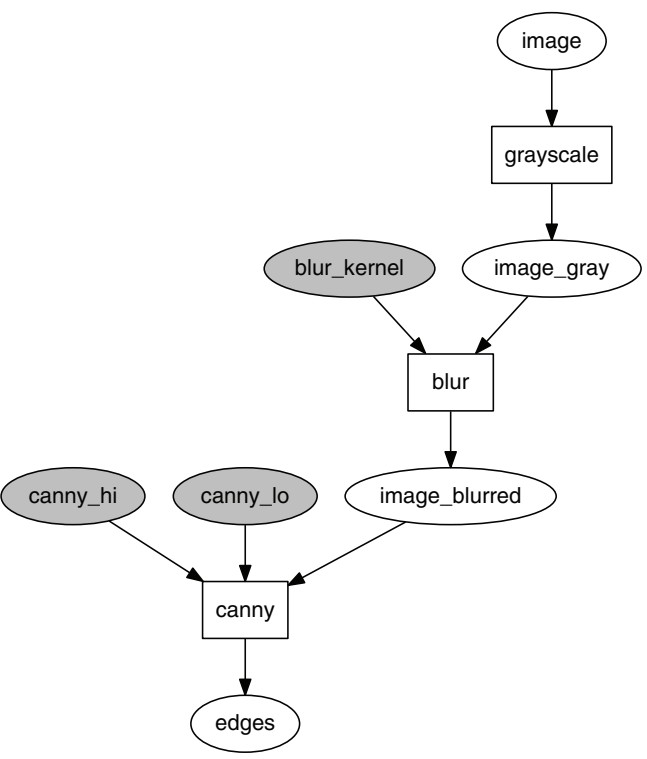

**Figure 4 Computational graph for edge detection.**

For example, to visualize the blurred image from the computational graph in Fig. 4 using Matplotlib, the following syntax is applied:

```
plt.imshow(runner['image_blurred'])
```

## Pipelines

To introduce additional functionality to algorithms expressed as computational graphs and transform them into runtime reactive components, a hierarchy of *pipeline* classes is defined.

As shown in Fig. 5, the basic building block of EPypes pipelines is a `Node`, which is a runtime counterpart to a function. An instance of `Node` based on function *f* can be invoked as a callable object, with parameter values corresponding to the positional input arguments of *f*. A network of node instances corresponding to the graph *G* form a `NodeBasedCompGraph`. The latter constitutes the main component of a `Pipeline`, as well as its subclasses (`SourcePipeline`, `SinkPipeline`, and `FullPipeline`).

An instance of the `Pipeline` class is constructed similarly to the earlier example of `CompGraphRunner`, but with the additional name argument:

```
pipe = Pipeline('MyPipeline', cg, hparams)
```

Because `Pipeline` is defined as a subclass of `Node`, its instances constitute callable objects, and are functionally equivalent to instances of `Node`. The whole pipeline is orchestrated by an instance of `CompGraphRunner` (Fig. 5). The internal structure of a `Pipeline` is visualized in Fig. 6.

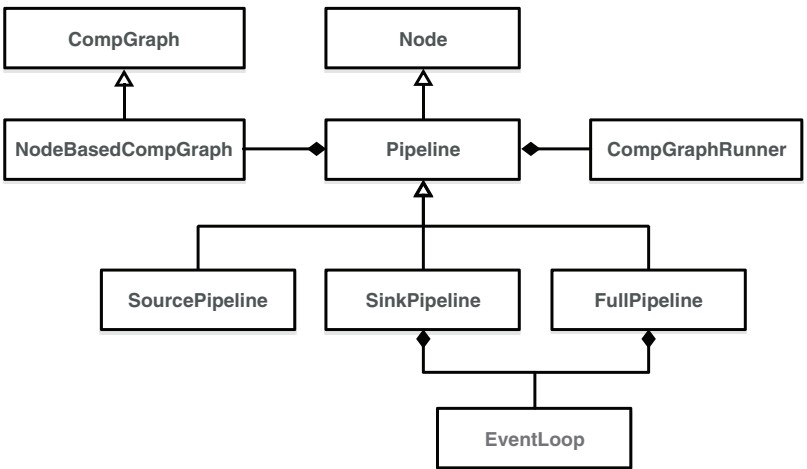

**Figure 5** **Class digram of EPypes Pipelines.**

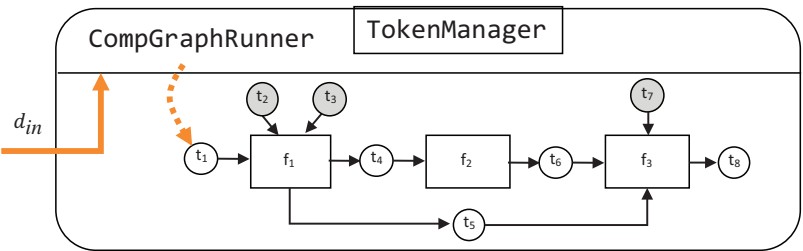

**Figure 6** **Structure of an instance of Pipeline.**

Additional capabilities of a `Pipeline`, as compared with a raw `CompGraphRunner`, include time measurement of nodes' durations, computation of computational graph overhead, storage of additional attributes, and other functionality added by subclassing `Pipeline`.

To allow for reactive behavior of pipelines, they are combined with event queues, which can be used for subscription to triggering events and publishing the results of data processing. To realize this, aside from `Pipeline`, which is not reactive, three other types of pipelines, coupled with event queues, are defined. Reactive pipelines operate in context of *thread-based concurrency* with blocking queues as the synchronization mechanism. In the Python standard library, a `queue.Queue` object can be used to communicate between two threads: the *producer* thread puts an object on the queue, and the *consumer* thread request the object and blocks until the latter becomes available. The principle of such interaction is shown in a sequence diagram in Fig. 7.

A `SourcePipeline`, see Fig. 8, is a subclass of `Pipeline` whose final output is put to the output queue $q_{\text{out}}$. A `SourcePipeline` is in addition parametrized by $f_{\text{out}}$, an output preparation function, responsible for packaging the chosen data from the pipeline tokens into a single message that gets published on $q_{\text{out}}$.

An instance of `SourcePipeline` is constructed as follows:

```
src_pipe = SourcePipeline('MySourcePipeline', cg, q_out, f_out, hparams)
```

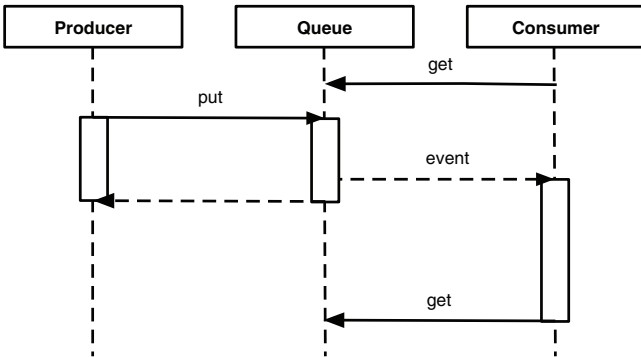

**Figure 7  Sequence diagram of thread-based producer and consumer interacting through a queue.**

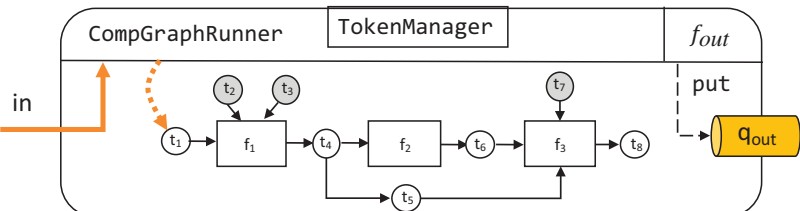

**Figure 8  Structure of an instance of `SourcePipeline`.**

As an example of the output preparation function, consider a pipeline, whose computational graph contains a token with the key `pose`, corresponding to a 3D pose estimated from images. To take the data corresponding to this token and package it as a Python pickle, the following function can be defined:

```python
def prepare_output(pipe):
    pose = pipe['pose']
    wire_data = pickle.dumps(pose)
    return wire_data
```

Another subclass of `Pipeline` is `SinkPipeline`, shown in Fig. 9. It is meant not to be called manually, but to be triggered as an event $e$ is announced in $q_{in}$. Because $e$ can be an arbitrary object, it is necessary to map its contents to a dictionary $d_e$ that describes what data should correspond to the pipeline's free source tokens. Such mapping is defined by event dispatcher function $f_{in}$.

An instance of `SinkPipeline` is constructed in a familiar way:

```python
snk_pipe = SinkPipeline('MySinkPipeline', cg, q_in, f_in, hparams)
```

The idea of event dispatcher can be illustrated by referring to the computational graph in the earlier example (Fig. 4). Consider that $e$ constitutes an image as a `numpy.ndarray`. Because a `CompGraphRunner` is invoked with keyword arguments, $f_{in}$ is defined to map to the required `kwargs` dictionary:

```python
def dispatch_image(im):
    return {'image': im}
```

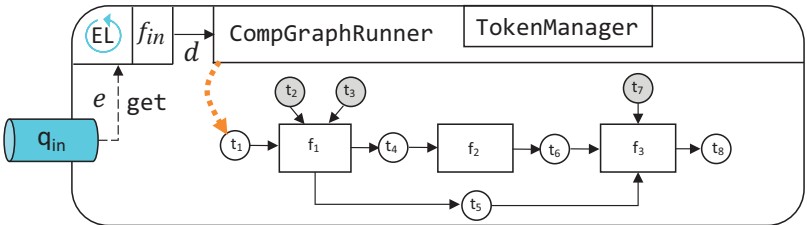

**Figure 9 Structure of an instance of `SinkPipeline`.**

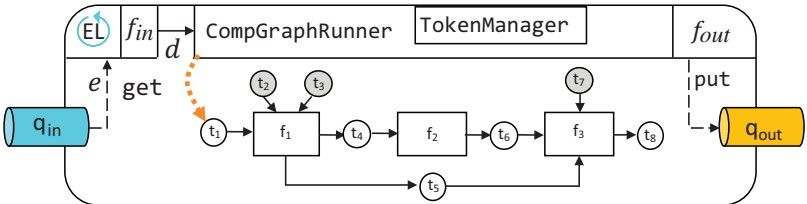

**Figure 10 Structure of an instance of `FullPipeline`.**

The behavior of waiting for an event is realized with an event loop, an instance of `EventLoop` class, which is continuously run in a separate thread of execution. It monitors $q_{in}$, and, as a new event $e$ becomes available, invokes the associated instance of `SinkPipeline` (Fig. 9) having the `kwargs` from the event dispatcher:

```
input_kwargs = self._event_dispatcher(event)
self._callback_pipeline.run(**input_kwargs)
```

Finally, the most comprehensive EPypes entity is `FullPipeline`, shown in Fig. 10. It subclasses `Pipeline`, and provides functionality of both reacting to a stream of incoming events in $q_{in}$ and publishing a subset of its processing results to $q_{out}$ as outgoing events. It is instantiated in the following way:

```
snk_pipe = FullPipeline('MyFullPipeline', cg, q_in, q_out, f_in, f_out,
hparams)
```

### EPypes-based system development

A distinction between a static computational graph and its runtime counterparts is realized in order to facilitate smooth system evolution from an early ad hoc development phase to a more integral whole with well-functioning reactive behavior. As shown in Fig. 11, the development starts with components having less structure, and proceeds by extension of these components with functionality and behavior that are facilitated by the proposed tools.

In the early development phase, vision algorithms, as well as other data processing routines, are prototyped using the available tool set: different alternatives can be implemented and evaluated in an interactive manner using tools like Jupyter and supported by OpenCV and a pool of scientific Python libraries (NumPy, Pandas, Scikit-image, Scikit-learn). As the result of prototyping, a collection of well-tested functions is developed. At this stage, the developer can specify computational graphs from the pool of these functions.

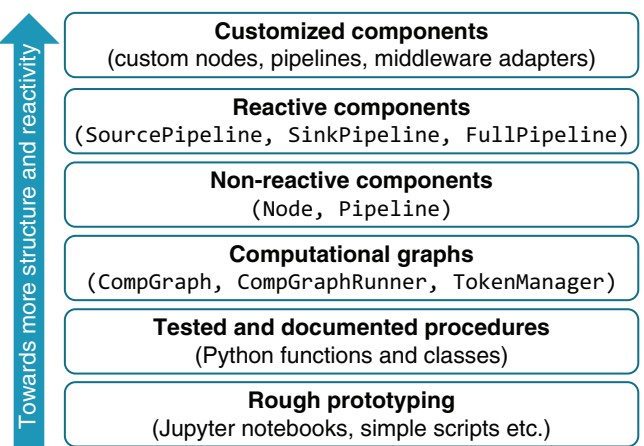

**Figure 11 Layered system development framework.**

The process of computational graph engineering involves a great deal of prototyping itself. Despite the fact that `CompGraph` constitutes a highly-structured entity, the flexibility of its definition brings a number of advantages over coding up the algorithm as a single function. Most importantly, the flat structure of the computational graph, along with Graphviz-based visualization capabilities, gives a transparent view on the data flow in the developed algorithm. It also allows for incorporating several alternative branches as a part of the same graph. The uniquely-named tokens provide an isolated namespace, which is specifically useful when prototyping in a Jupyter notebook. The mechanism of hyperparameter tokens allows for systematic management of the set of thresholds and other configuration values while being on a single hierarchical level (without a cascade of function calls). The well-defined structure of a computational graph facilitates automated manipulation of it, for example, extending the original graph with additional functions, union of two or more graphs, and union with renaming of functions and tokens.

When a computational graph is developed, it can be used to construct pipelines. The non-reactive `Pipeline` provides additional capabilities to the computational graph: it is runnable, includes time measurement functionality, and can be flexibly subclassed, as done in reactive pipelines (`SinkPipeline`, `SourcePipeline`, and `FullPipeline`). The latter are used to expose the developed algorithm in online mode.

## EPypes use case

In order to illustrate practical application of the EPypes framework and show its suitability for building data processing components in distributed environments, this section presents a run time use case scenario with the associated experiment. The presented scenario demonstrates how EPypes can be deployed as a part of a real distributed system (with the communication based on ZeroMQ and Protobuf) and what timing properties may be expected in this case. In particular, a major concern is how much overhead is introduced by the additional abstractions in the EPypes architecture. Furthermore, it is of interest how repeatable this overhead is, as well as what role it plays comparing to communication latency and the application-specific processing time.

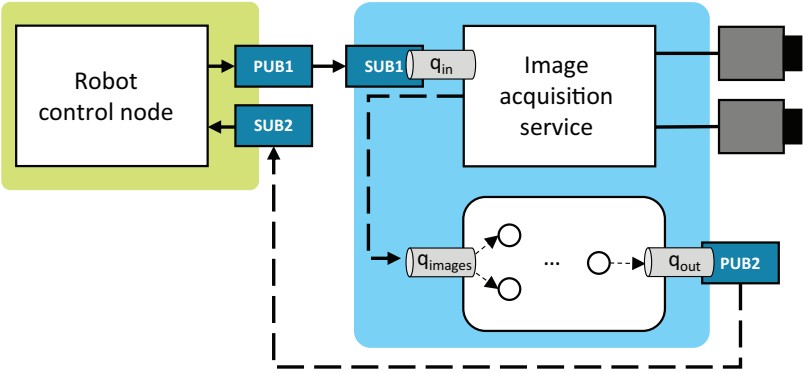

**Figure 12 System configuration.**

## System description

As shown in Fig. 12, the case system is comprised of three nodes: (1) the robot control node, (2) the image acquisition service, and (3) the EPypes-based image processing node. The robot control node coordinates the robot's work cycle and realizes communication with external systems. The system performing stereo acquisition from two cameras is designed as a streaming service, built using the FxIS framework (*Semeniuta & Falkman, 2018*). For each associated camera, a stream of images is captured in its own thread of execution, and a number of recent frames are retained at each moment. External systems can request images from the service that closely correspond to the request timestamp.

The nodes run in the distributed environment and communicate through ZeroMQ publish/subscribe sockets and in-process blocking queues. For publishing and subscribing, EPypes provides two thread-based abstractions, namely `ZMQPublisher` and `ZMQSubscriber`. The former encapsulates a ZeroMQ PUB socket and acts as a consumer of an in-process queue: as a new data is available on the queue, it gets published. An example in Fig. 12 is the PUB2/$q_{out}$ pair. `ZMQSubscriber` encapsulates a ZeroMQ SUB socket, which is polled with the `Poller` object. On arrival of a new message, the latter is put on the connected in-process queue. An example in Fig. 12 is the SUB1/$q_{in}$ pair

The robot control node runs on an ARM-based Raspberry Pi 3 single-board computer with the Raspbian operating system, while the vision-related components are deployed to an Ubuntu-based x86-64 machine. The latter has an Ethernet connection to a stereo camera pair (GigE Vision-based Prosilica GC1350), which are used by the image acquisition node.

The following communication loop is considered:

1. Robot announces request for timely image processing results; the image request is announced asynchronously as an event at PUB1.
2. Images most closely associated with the request are acquired and, as a tuple of `numpy. ndarray`, communicated to the processing component via the common in-process queue $q_{images}$.
3. Image processing node extracts the desired features from the images, which are communicated back to the robot via the PUB2/SUB2 asynchronous socket pair.
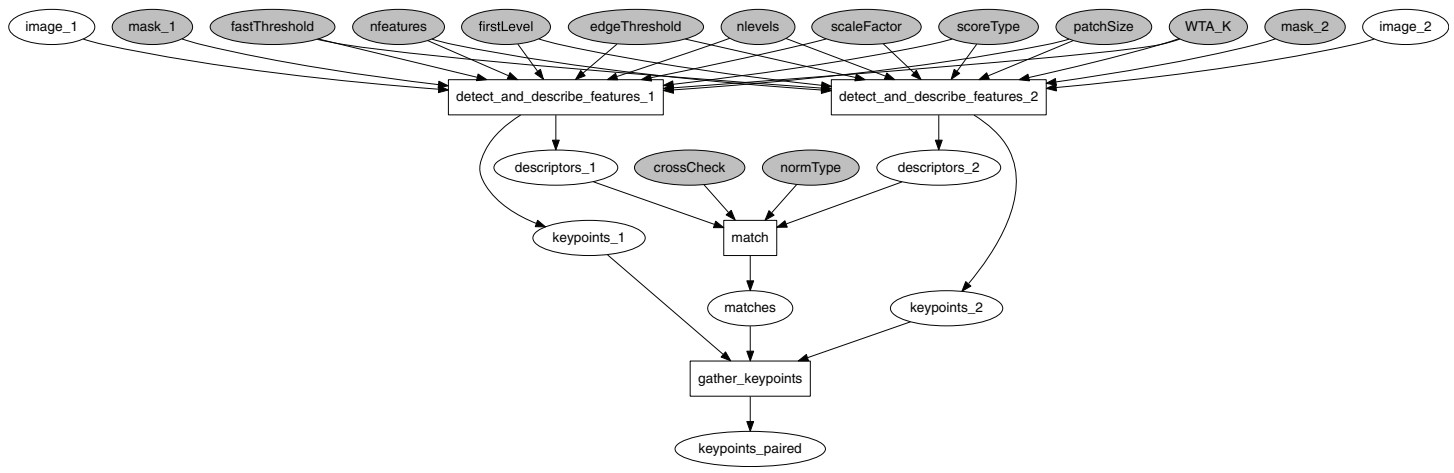

**Figure 13 ORB computational graph.**

The target vision algorithm performs ORB feature detection, description, and matching (*Rublee & Bradski, 2011*). Figure 13 shows the corresponding computational graph. After image features are identified in each image, collections of feature descriptors are matched against each other using OpenCV's `BFMatcher` object, with the matches returned in sorted order by match distance. The final `gather_keypoints` function produces an array of the matched keypoints' coordinates.

The communicated messages that are send over wire are serialized in the Google's Protocol Buffers (Protobuf) interchange format. Three message types are used:

- `AttributeList` represents a collection of key/value attributes, where an attribute can be either a `string`, a `double`, or an `int32`.
- `Event`, sent over PUB1/SUB1, is comprised of an `id` (`string`), a `type` (`string`), and `attributes` (`AttributeList`);
- `JustBytes`, sent over PUB2/SUB2, is comprised of an `id` (`string`), `content` (`bytes`), and `attributes` (`AttributeList`);

The computational graph shown in Fig. 13 forms a basis for an instance of a `FullPipeline`. Its event dispatcher $f_{in}$ handles tuples with pairs of images put onto $q_{images}$. The output preparation function $f_{out}$ is responsible for packaging the output data as a `JustBytes` Protobuf message, with its `content` being the Pickle-serialized value of the first 20 rows of the `keypoints_paired` token (`numpy.ndarray`), and the `attributes` filled by timestamps and durations captured with the image acquisition service and the EPypes pipeline.

## Time measurement experiment

The robot control node announces a series of vision requests and extracts attributes from the response Protobuf messages. In addition, it records the timestamps of when the vision request get announced ($t_{vreq}$) and when the corresponding response is obtained ($t_{vresp}$).

The difference between these timestamps accounts for the trip duration of the current request:

$$\tau_{\text{trip}} = t_{\text{vresp}} - t_{\text{vreq}}$$

For execution of both the image acquisition service and the vision pipeline, two timestamps are added to the properties set: $t_{\text{react}}$, when the component reacted to the incoming event, and $t_{\text{pub}}$, right before publishing the outgoing event. Their difference $\tau_{\text{r}\to\text{p}}$ provides the measurement of the component's processing time, including processing of incoming and outgoing events:

$$\tau_{\text{r}\to\text{p}} = t_{\text{pub}} - t_{\text{react}}$$

Properties related to the vision pipeline that get added to the outgoing Protobuf message comprise vision processing time $\tau_p$, overhead from orchestrating the computational graph $o_{\text{cg}}$, and timestamps of start and finish of the event dispatcher $f_{\text{in}}\left(t_{f_{\text{in}}\uparrow}, t_{f_{\text{in}}\downarrow}\right)$ and the output preparation function $f_{\text{out}}\left(t_{f_{\text{out}}\uparrow}, t_{f_{\text{out}}\downarrow}\right)$, which define the corresponding function durations:

$$\tau_{f_{\text{in}}} = t_{f_{\text{in}}\downarrow} - t_{f_{\text{in}}\uparrow}$$
$$\tau_{f_{\text{out}}} = t_{f_{\text{out}}\downarrow} - t_{f_{\text{out}}\uparrow}$$

Computational graph overhead $o_{\text{cg}}$ is measured internally by the pipeline (`p.compute_overhead()`), and constitutes the difference between total processing time of the pipeline and the sum of processing times of all the enclosed nodes:

$$o_{cg} = \tau_p - - \sum \{\tau_n \text{ for each node } n \in p\}$$

After each request is finished, the robot control node records all the obtained properties. The latter are further aggregated in a Pandas data frame, with a row of properties' values per each request. From the available data, the following overhead metrics can be computed:

1. *Network overhead* measures how much the trip duration is greater than the time spent in all the components:

$$o_{\text{network}} = \tau_{\text{trip}} - \left(\tau_{r\to p}^{(image\ acquistition)} + \tau_{r\to p}^{(vision\ pipeline)}\right)$$

2. *EPypes overhead* is computed as an excess time in the vision pipeline in addition to the processing in the computational graph and in the functions $f_{\text{in}}$ and $f_{\text{out}}$:

$$o_{\text{epypes}} = \tau_{r\to p}^{(vision\ pipeline)} - \left(\tau_p + \tau_{f_{\text{in}}} + \tau_{f_{\text{out}}}\right)$$

Figure 14 demonstrates the timeline of 100 vision requests and the associated durations of the image acquisition service, the vision pipeline, and the overhead from network communication.

Data has been collected from five experiments, each with 500 vision requests. For each experiment, a maximum likelihood estimation of log-normal probability density function is performed for distributions of $o_{\text{cg}}$ and $o_{\text{epypes}}$. The same estimation is performed for all data combined. Figures 15 and 16 show visualization of these PDFs. A PDF for each individual experiment is visualized as a shaded area under the curve. The PDF for all data is shown as a thick curve. The thin vertical line specify the

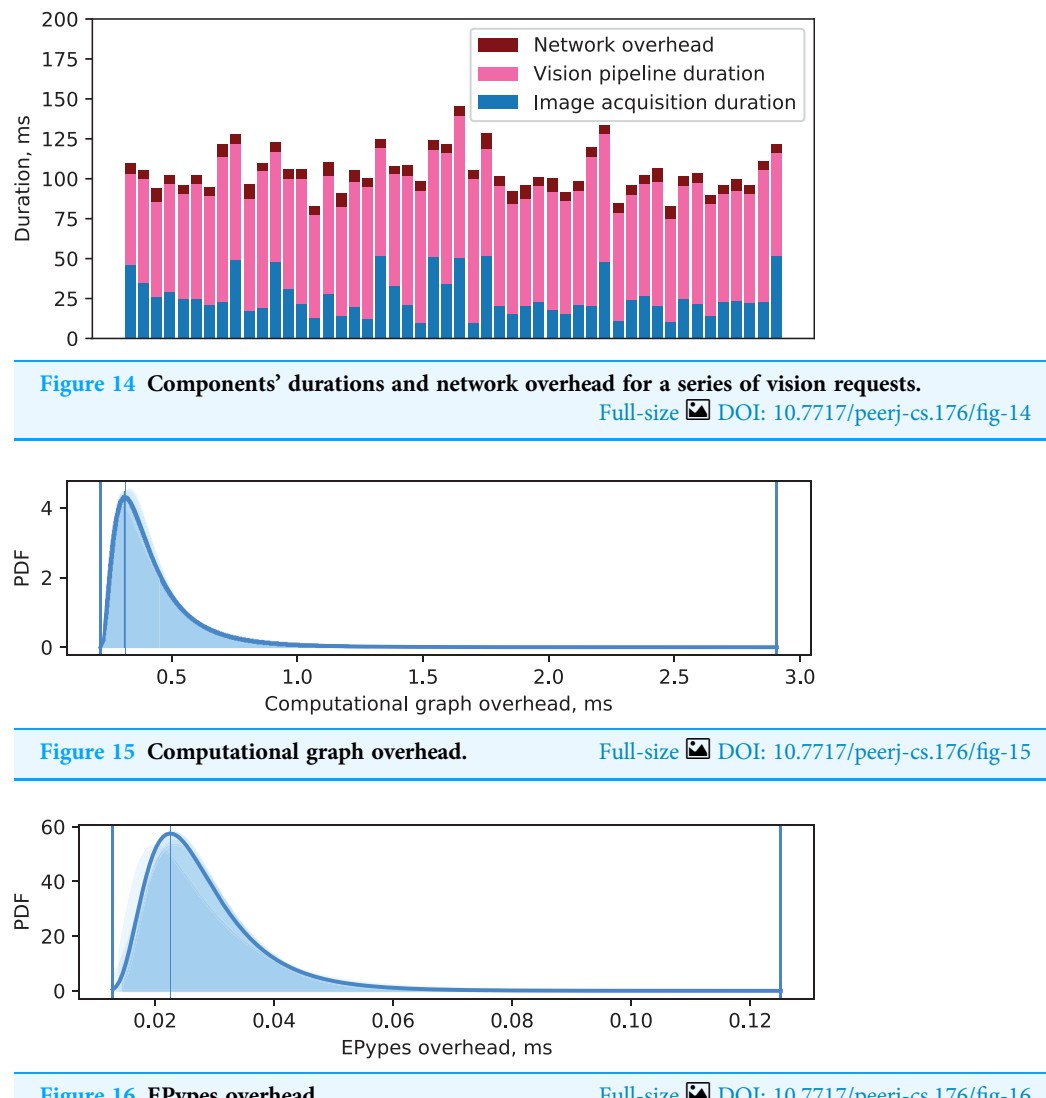

**Figure 14 Components' durations and network overhead for a series of vision requests.**

**Figure 15 Computational graph overhead.**

**Figure 16 EPypes overhead.**

modal value of the PDF for the combined dataset, and the enclosing thick vertical lines delimit the overall range of measurements for the combined dataset.

It can be seen from Fig. 15 that overhead from performing data processing based on a computational graph $o_{cg}$ is characterized by matching log-normal distributions for every experiment, with most of the probability density located around 0.3 ms. The EPypes overhead $o_{epypes}$, as shown in Fig. 16, has much tighter range of possible values, distributed log-normally with matching distributions for every experiment, and with most of the probability density around 0.02 ms. Overall, for a vision algorithm that naturally requires tens of milliseconds to perform the processing, the overheads introduces by EPypes can be considered negligible.

## Related work

The idea of explicit utilization graph-based representation of data processing algorithms has been on the surface for many years. The availability of engineering tools, data science

frameworks, and modeling formalisms, described in the Background section, shows the efficacy of the pipeline thinking when designing systems with streaming logic. The distinctive approach of EPypes lies in its tight integration with the Python ecosystem, support for algorithm prototyping, and abstractions for integration of the developed computational graphs into distributed systems.

The EPypes architecture is a logical continuation of the concept of discrete event data flow, earlier presented by *Semeniuta & Falkman (2015)*. This earlier work attempted to define a data flow formalism with distinct notion of event as the one used in publish/subscribe systems. However, the presented formalism didn't include a reference implementation at the time. EPypes has, in turn, refined the notion of reactive pipelines and made it usable in real scenarios.

Other highly related work within the formal methods domain is Stream Algebra (*Helala, Pu & Qureshi, 2014*), with its Go-based implementation (*Helala, Pu & Qureshi, 2016*). This approach models an image processing algorithm as a set of data streams that get altered by a set of operators. In the algebra implementation, a stream corresponds to a Go channel, and the set of defined operators allow to define usable workflow patterns such as pipeline graphs, fork-join graphs, and pipeline graphs with feedback. The latter option is naturally supported due to the concurrency features of Go. This approach, similarly to EPypes, allows to construct high level algorithm from finer functions, including those from the OpenCV library. The distinctive feature is the support for feedback, which is disallowed in EPypes due to the acyclicity requirement. The feedback with EPypes, however, can be realized on a higher systemic level, by incorporating additional distributed components.

In the contemporary robotics research, the robot operating system (ROS) is widely used as the underlying platform for the distributed robotic applications relying on data from sensors and cameras. The general architecture in this case is based on a collection of *nodes* that react to arrival of data through publish/subscribe topics, which makes the overall logic graph-based. The related concept of *nodelet* (and *component* in ROS2) allows to realize a processing graph structure as a part of a single operating system process. Examples of this approach is often demonstrated on the applications of point cloud processing (*Rusu & Cousins, 2011*; *Munaro et al., 2013*; *Carraro, Munaro & Menegatti, 2017*), as to minimize latency due to inter-process or remote communication. ROS-based processing graphs, especially in the single-process case, are somewhat similar to EPypes pipelines. They, however, target applications with already developed algorithms, as opposed to EPypes, which supports early-stage prototyping using the graph-based abstractions.

Other academic examples of similar robot/vision architectures include the one based on the supervisory control theory of discrete-event systems (*Košecka, Christensen & Bajcsy, 1995*) and service-oriented dataflow-like components, auto-tuned by higher-level supervisors (*Crowley, Hall & Emonet, 2007*).

## CONCLUSIONS AND FURTHER WORK

This paper has presented EPypes, an architecture and Python-based software framework for building event-driven data processing pipelines. Because most of vision

algorithms and many data processing routines are naturally modeled as pipelines, EPypes offers a capability of implementing data processing systems as DAGs. Apart from the functional components comprising the prototype implementation of EPypes, this paper has presented a system development framework that supports evolution of computational graphs from an early prototyping phase to their deployment as reactive pipelines.

The principle of the EPypes abstraction is demonstrated on the example of constructing a computational graph for edge detection and discussing the inner structure of the hierarchy of pipelines. Further, a real scenario of deployment of an EPypes pipeline for features detection and matching to a distributed system is experimentally studied. It was shown that the ability to adapt reactive behavior to various publish/subscribe middleware solutions allows to combine EPypes pipelines with already available systems. The measured timing properties of the image processing component based on EPypes show that the latter introduces negligible overhead comparing to the application-inherent processing time.

An important part of further work should be connected with development of software abstractions on the highest level of the system development continuum shown in Fig. 11. This will enable fine-tuning and enhancing of reactive pipelines, for example, with adapters to different messaging systems (e.g., MQTT, RabbitMQ, DDS), parallelizable nodes, and specialized pipeline management logic. An important task in this case is implementation of systematic error handling. A failure inside the pipeline (e.g., in the case of a vision system, due to changed lighting conditions) can be handled by issuing the corresponding event that will be processed by a remote component. In addition to queues providing asynchronous messaging, other communication modalities can be used. An RPC API (such as REST or gRPC) can be established to allow external systems getting meta-information about the running pipeline and changing values of hyperparameters. Last, but not least, functionality for interaction with databases should be integrated.

As the presented software framework is implemented in Python, it naturally gears toward system prototyping use cases. The static abstractions are useful for algorithm prototyping, while the transition to the reactive components allow for rapid deployment of the computational graphs to distributed environments. This allows for harnessing the available Python data science tools and integrating them into industrial automation workflow.

The limitation of the proposed implementation lies in its non-deterministic overhead due to the use of the interpreted garbage-collected programming language. Hence, applications requiring high rate of operation and more deterministic running time are more likely to be developed in C++ with custom UDP-based communication protocols or real-time middleware such as DDS. It is of interest therefore to validate the principles of EPypes using C++ codebase, as well as to devise a strategy of transforming EPypes-based computational graphs to high-performance computing components, for example, via code generation.

### Funding

This paper was written in association with the MultiMat project and SFI Manufacturing, funded by the Norwegian Research Council. The funders had no role in study design, data collection and analysis, decision to publish, or preparation of the manuscript.

### Grant Disclosure

The following grant information was disclosed by the authors:
MultiMat project and SFI Manufacturing, funded by the Norwegian Research Council.

### Competing Interests

The authors declare that they have no competing interests.

### Author Contributions

- Oleksandr Semeniuta conceived and designed the experiments, performed the experiments, analyzed the data, contributed reagents/materials/analysis tools, prepared figures and/or tables, performed the computation work, authored or reviewed drafts of the paper, approved the final draft.
- Petter Falkman authored or reviewed drafts of the paper, approved the final draft.

### Data Availability

The source code of the EPypes library is available at GitHub:

https://github.com/semeniuta/EPypes.

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
