# Peer review of "EPypes: a framework for building event-driven data processing pipelines"

_PeerJ Computer Science, doi:10.7717/peerj-cs.176_

## Round 0.1 · original submission · Major Revisions

Please follow carefully the hints of the two anonymous reviewers.
Specifically, to avoid reject, pay particular attention to the substantial weakness in the approach assessment (experiment and validity of findings) since these problems were highlighted by both reviewers.

Reviewer 1 ·

Basic reporting

The paper is in general nicely written, with good use of the English language in a professional manner. However, there are some typos, unclear statements, and formatting that need to be addressed.

Line number: Comment
18: Development -> Developing?
20: To -> to, this paper -> This paper
35: Communication fabric? What does this mean?
38: tools of trade -> tools of the trade
70: Python -> a/the Python
77: only this -> only is this
94: directed acyclic graphs -> directed acyclic graphs (DAG).
101: movement gained -> movement has gained
115: In distributed systems -> In distributed systems,
135: computational graphs' execution -> execution of computational graphs. See also line 79.
140: i.e. -> i.e.,
141: to much space above and below the equation.
142: the word where should not start a new paragraph.
146: indegree and outdegree. rephrase.
148-149: Ugly formatting of In_f and Out_f. Please fix.
153: conform -> conforms
166: It is not really clear what the token sets represent. Maybe reformat and put in some more context.
197: Don't start a sentence with a symbol (func_io).
228: operation the same -> operation is the same.
240: of Pipeline -> of the Pipeline
253: as synchronization -> as the synchronization
264: SE(3) is not introduced or defined. Rephrase.
284: continuously-run -> continuously run
301: available tool set? which tools? rephrase.
400: Hard to read. rephrase.

The paper follows a semi-traditional structure. The related work section is presented after the main body of the text. This is in itself not a problem. However, the related work section is too short and needs to be expanded and the presented work must in a much greater detail be compared to other systems/frameworks. E.g., in robotics, ROS is much used for vision prototyping and development. How does this work relate to ROS nodes, nodelets, etc.?

The figures in the paper are very good.

The paper is self-contained, but needs to be expanded. See comment above.

Experimental design

The problem description presented in lines 57-63 is good.

The experimental results are severely lacking. It is not clear what the experiment presented in section EPYPES USE CASE is meant to convey. The timing results are not placed in context and discussed.

Validity of the findings

The conclusion is written like a summary. Please connect the conclusion to the problem description presented in the introduction. Important findings from the experimental results should be included.

·

Basic reporting

- I checked the code and it works well

Experimental design

- The python code should not include inside the text. The authors maybe can replace with algorithms or pseudo code.
- The study contains some similar descriptions and figures (Fig. 6, 7, 9, 10) I am wondering whether a way to merge some of them to make the comparison clearly.
- References are weak, the authors might need to make their work stronger with more related references.

Validity of the findings

- How their work is different from previous work (Semeniuta, 2015). How much better performance is there? What is the statistical significance of improvement? It needs more discussions.
- Authors should compare with other related works.

Additional comments

Although the authors had done a good work with, I recommend the authors should organize their methods and results in a better way. Now it contains a lot of explanations, codes, and figures that make readers be hard to follow. Also, please discuss more your results to make your contributions significantly.

---

## Round 0.2 · Minor Revisions

Both reviewers re-reviewed. One reviewer suggested performing carefully proofreading correcting typos that are still present in the manuscript.

After that, the paper is ready for publication.

Reviewer 1 ·

Basic reporting

There are some typos here and there, e.g., UPD instead of UDP in the conclusion, that should be fixed.

The paper is substantially improved.

Experimental design

no comment

Validity of the findings

no comments

·

Basic reporting

No comment

Experimental design

No comment

Validity of the findings

No comment

---

## Round 0.3 · accepted · Accept

As a final remark, before the final submission, authors are invited to perform a last careful proofread to fix typing errors that may still be present in the manuscript to ensure the highest quality manuscript for PeerJ CS.

Thank you!